# Bardet–Biedl Syndrome—Multiple Kaleidoscope Images: Insight into Mechanisms of Genotype–Phenotype Correlations

**DOI:** 10.3390/genes12091353

**Published:** 2021-08-29

**Authors:** Laura Florea, Lavinia Caba, Eusebiu Vlad Gorduza

**Affiliations:** 1Department of Nephrology-Internal Medicine, Faculty of Medicine, “Grigore T. Popa” University of Medicine and Pharmacy, 16 University Street, 700115 Iasi, Romania; laura.florea@umfiasi.ro; 2Department of Medical Genetics, Faculty of Medicine, “Grigore T. Popa” University of Medicine and Pharmacy, 16 University Street, 700115 Iasi, Romania; eusebiu.gorduza@umfiasi.ro

**Keywords:** Bardet–Biedl Syndrome, ciliopathy, heterogeneity, pleiotropy, variable expressivity

## Abstract

Bardet–Biedl Syndrome is a rare non-motile primary ciliopathy with multisystem involvement and autosomal recessive inheritance. The clinical picture is extremely polymorphic. The main clinical features are retinal cone-rod dystrophy, central obesity, postaxial polydactyly, cognitive impairment, hypogonadism and genitourinary abnormalities, and kidney disease. It is caused by various types of mutations, mainly in genes encoding BBSome proteins, chaperonins, and IFT complex. Variable expressivity and pleiotropy are correlated with the existence of multiple genes and variants modifiers. This review is focused on the phenomena of heterogeneity (locus, allelic, mutational, and clinical) in Bardet–Biedl Syndrome, its mechanisms, and importance in early diagnosis and proper management.

## 1. Introduction

Ciliopathies are diseases caused by the dysfunction of motile and non-motile primary cilium [1]. Primary cilia are involved in numerous cellular processes. Such as cell cycle control, development, migration, polarity, differentiation, stimuli transduction, proliferation, and maintenance of stem cells [2,3,4]. Primary cilia also intervenes in the cellular signaling pathways of development and homeostasis: hedgehog, Wnt (wingless-related integration site), Notch, Hippo, GPCR (G protein-coupled receptors), PDGF (platelet-derived growth factor), mTOR (mammalian target of rapamycin), and TGF-βeta (βeta transforming growth factor-β) [5,6].

Cilia are microtubule-based organelles. They are anchored to the cytoskeleton and protrude at the cell surface [7]. They have three main components: axoneme, basal body and transition zone. The axoneme consists of nine peripheral microtubule doublets, and a central part that could be formed by a pair of microtubules (model 9 + 2) or by the absence of this pair of microtubules (model 9 + 0) [8]. The transport in and out of cilia is allowed by three complexes: intra-flagella transport (IFT) complex A (IFT-A), IFT complex B (IFT-B) and BBSome [4]. IFT-A is involved in the retrograde transport (tip to the base), IFT-B is involved in the anterograde transport (base to the tip) while BBSome is a multiprotein complex implied in ciliary trafficking activity [9].

Ciliopathies are characterized by a high clinical and molecular heterogeneity and a large clinical overlap between entities [10]. The clinical expression of the cilia dysfunction is correlated with the activity of cilia. Motile cilia dysfunction causes hydrocephalus, infertility, chronic respiratory issues, but also congenital heart defects and organ laterality defects (the last two are common manifestations of non-motile cilia dysfunction) [11]. Non-motile cilia dysfunction determines: retinal dystrophy, anosmia, hearing loss, central obesity, skeletal abnormalities (polydactyly, rib cage), hypogonadism, genital abnormalities, ataxia, epilepsy, mental disability, brain malformations, facial abnormalities, renal abnormalities (polycystic kidney disease-PKD, nephronophthisis-NPHP), and liver disease (liver fibrosis) [11].

Bardet–Biedl syndrome (BBS, OMIM 209900) is a rare autosomal recessive multisystem non-motile ciliopathy primarily characterized by heterogeneous clinical manifestations. The prevalence of BBS is high in inbred/consanguineous populations. In the general population, BBS has a prevalence of 0.7/100,000 and a prevalence at birth of 0.5/100,000 [12]. However, the incidence of BBS is variable: 1 in 160,000 in North America and Europe, 1 in 17,000 in Kuwait—Bedouin populations; 1 in 3700 individuals in the Faroe Islands [13,14,15,16].

## 2. Pleiotropy and Variable Expressivity in Bardet–Biedl Syndrome

BBS is characterized by a high genetic heterogeneity (locus, mutational, clinical), variable expressivity and pleiotropy. There are 24 loci involved and the mutational profile is diverse. There are modifying variants and numerous interactions between the BBS proteins (interactions disease protein—disease protein) or between the BBS protein and another protein (interactions disease protein—non disease protein). These explain the extremely polymorphic clinical picture that includes both major and minor features (Figure 1).

This pleiotropic disorder has a constellation of features, which are divided into major features (rod-cone dystrophy, central obesity, postaxial polydactyly, hypogonadotropic hypogonadism and/or genitourinary abnormalities, cognitive impairment, kidney disease) and minor features (developmental delay, speech deficit, brachydactyly or syndactyly, dental defects, ataxia or poor coordination, olfactory deficit, diabetes mellitus, and congenital heart disease [17,18,19,20].

Before the discovery of BBS genes, Beales et al. established a diagnostic algorithm based on phenotypic presentations of this syndrome [20].

### 2.1. Major Features

One of the most important features of BBS is the rod-cone dystrophy, affecting 94–100% of individuals [21,22]. The patients can present night blindness, peripheral vision loss, diminution of color, and overall loss of visual acuity [23]. Rod and cone photoreceptor cells are modified ciliated cells responsible for the night and day vision. The main protein of rod cells is rhodopsin. The transport of this protein from the inside to the outside of rod cells is allowed by BBSome complex. Mutations in BBSome genes lead to mislocalization and accumulation of rhodopsin in rod cells. In this way the cellular homeostasis is disturbed and the degeneration of the photoreceptors is produced [24].

Obesity is a frequent feature (89%), it affects the thorax and abdomen and has an early onset (from the age of 2–3 years) [22]. Obesity is produced by dysregulation of appetite, altered leptin resistance, impaired leptin receptor signaling, altered neuroendocrine signaling from ciliated neurons to fat storage tissues, reduced number of cilia due to BBS gene mutations and alteration ways involved in the differentiating preadipocytes [25]. Common adult obesity was associated with nucleotide polymorphism in *BBS2* (rs4784675). Early onset childhood obesity and common adult morbid obesity was associated with *BBS4* (rs7178130) and *BBS6* (rs6108572 and rs221667) changes [26].

Postaxial polydactyly is frequently reported in 79% of patients, with polydactyly of toes being more common than the polydactyly of fingers [22]. The gene *BBS17*, which is a negative regulator for the ciliary trafficking mediated by BBSome and the Shh (Sonic hedgehog) signaling, was associated with mesoaxial polydactyly [27].

Cognitive impairment is present in 66% cases with BBS [22]. Patients can present either intellectual disability or impairments in verbal fluency, attention capacity, poor reasoning, and emotional immaturity. The connection between BBS proteins and cognitive impairments remains unknown [22].

Hypogonadism and genitourinary malformations have an incidence of 59% [22]. Hypogonadism may be apparent at puberty. Male anomalies can vary from micro penis, small-volume testes, cryptorchidism to hypogonadotropic hypogonadism [28]. Female anatomic anomalies are uterine, fallopian, ovarian or vaginal hypoplasia, or atresia [22]. The rate of fertility is low, but both sexes can have biological children [20].

Kidney disease is present in 52% of BBS patients [22]. The spectrum of kidney disease varies from urinary tract malformations (vesicoureteral reflux, hydronephrosis, dysplastic cystic disease, absent, duplex, horseshoe or ectopic kidneys, neurogenic bladder) to chronic glomerulonephritis and defective tubular concentrating ability [18]. The main consequence is chronic kidney disease (CKD), which contributes to morbidity and mortality in patients with BBS. The majority of cases with renal disease is completely diagnosed at the age of five, but some features could be discovered in the first year of life [29]. Progression of CKD is increased by arterial hypertension or type 2 diabetes mellitus. End stage renal disease requires dialysis or a transplant. After a renal transplant, favorable long-term outcomes have been reported [30].

### 2.2. Minor Features

Minor features are represented by various anomalies in different systems and organs: the nervous system, sensorial changes, cardiovascular system, gastrointestinal changes, different endocrine glands, and cutaneous or musculoskeletal changes [20].

Neurodevelopmental abnormalities observed in patients with BBS include ataxia and poor coordination with mild hypertonia of all four extremities, seizures, speech abnormalities, behavioral, and psychiatric abnormalities [20].

The dysmorphism in BBS is not specific, and the most frequent reported features are: a narrow forehead, brachycephaly or macrocephaly, large ears, short, narrow, and downslanted palpebral fissures, deep and widely set eyes, malar flattening, depressed nasal bridge, long and smooth philtrum, retrognathia [20].

Anosmia/hyposmia are related to defects in olfactory cilia or the olfactory bulb [31]. Oral/dental abnormalities (50%), such as hypodontia or microdontia, high-arched palate, enamel hypoplasia, posterior crossbite, dental crowding, short roots, and taurodontism are reported [20].

Cardiovascular anomalies are relatively frequent (up to 29% of individuals) and are represented by atrioventricular septal defects, bilateral persistent superior vena cava, interrupted inferior vena cava, and hemiazygos vein continuation. Moreover, there are reports of laterality defects, such as situs inversus totalis, midline abdominal organs, asplenia, or polysplenia [22].

Gastrointestinal abnormalities can range from Hirschsprung disease, anatomic anomalies of the gastrointestinal tract (bifid epiglottis, laryngeal, and esophageal webs, bowel atresia, imperforate anus) to inflammatory bowel disease, celiac disease, and liver disease [32].

Endocrine/metabolic abnormalities include metabolic syndrome (54.3%), obesity, hyperlipidemia (usually hypertriglyceridemia), insulin resistance, and elevated fasting plasma glucose with or without type 2 diabetes mellitus (15.8%), polycystic ovarian syndrome (14.7%), and subclinical hypothyroidism (19.4%). Subclinical hypothyroidism is a mild form of hypothyroidism characterized by peripheral thyroid hormone levels within normal reference laboratory range but serum thyroid-stimulating hormone (TSH) levels are mildly elevated. Hashimoto’s thyroiditis is the most common form of subclinical hypothyroidism and is more prevalent in BBS patients compared to the general population [28,32,33].

In addition other features were reported: cutaneous dermatoses (seborrheic dermatitis, keratosis pilaris, striae, hidradenitis suppurativa, acanthosis nigricans), subclinical sensorineural hearing loss, musculoskeletal abnormalities (scoliosis, leg length discrepancy, club foot, Blount disease, and joint laxity) [20,32].

The clinical diagnosis of BBS is made in the presence of either four major features or three major features and two minor features [20]. To determine the genetic cause in BBS, gene-targeted testing (multigene panel) or comprehensive genomic testing (exome sequencing) is recommended. BBS-specific panels or larger ciliopathy genes panels can be used. In general single-gene testing by sequence analysis and deletion/duplication analysis is not recommended [32]. This genetic testing approach is based on the fact that BBS is characterized by locus and allelic heterogeneity, and because of the clinical overlap between ciliopathies [10].

## 3. Determinants of Clinical Effects

### 3.1. Locus Heterogeneity

In Table 1, there is a summary of data concerning the genes involved in BBS in correlation with tissue and single cell type specificity [34,35,36,37].

There are three categories of gene effects: primary effect (at the molecular level—protein), secondary effect (at the cellular level), and multiple tertiary effects (pleiotropic)—at the phenotypic level of organ/organism (signs and symptoms) [38]. Genes are expressed in all tissues, but there are differences in expression levels (Table 1).

The majority of genes involved in BBS presents a low tissue specificity. However, the *LZTFL1* gene is mainly expressed in lymphoid tissue while the *BBIP1* gene is specific to testis and the *NPHP1* gene has an expression in skeletal muscle (Table 1). For single cell type specificity, a cell type enhanced expression has been described in: ciliated cells (*BBS1*, *BBS2*, *ARL6*, *BBS4*, *WDPCP*, *LZTL1*, *IFT27*, *IFT74*, *NPHP1*), rod photoreceptor cells (*BBS1*, *ARL6*, *BBS4*, *BBS5*, *BBS7*, *TTC8*, *BBS9*, *BBS12*, *CEP290*, *NPHP1*, *SCAPER*), cone photoreceptor cells (*BBS1*, *ARL6*, *BBS4*, *BBS5*, *TTC8*, *BBS9*, *SCAPER*), spermatocytes (*ARL6*, *LZTFL1*, *BBIP1*, *IFT74*, *NPHP1*), early spermatids (*BBS5*, *BBS12*, *BBIP1*, *NPHP1*), late spermatids (*BBIP1*), alveolar cells type 1 (*WDPCP*), alveolar cells type 2 (*WDPCP*), club cells (*WDPCP*) [36].

Some BBS forms are considered chaperonopathies because three genes involved in BBS encode chaperon-like proteins. The three chaperonins-like proteins—MKKS, BBS12, BBS7—are involved in assembling BBSome, a multistep process [4,39,40]. Chaperonin like BBS proteins are involved in the first stage of assembly of the BBSome and mutations in genes that encode these proteins block the formation of functional complexes [4]. BBS caused by mutations in chaperonin genes has more severe forms characterized by an earlier onset (especially *BBS10*), higher prevalence of primary diagnostic signs and some borderline signs with other ciliopathies, such as McKusick–Kaufman syndrome (MKKS) and Alström syndrome [4,39,41]. This increased severity could be linked to residual activity/function gain of BBSome genes [4,42].

Chaperonin-like BBS genes are characterized by a small number of coding exons (one to four) and, thus, a mutational screening of these genes be applied before a more complex analysis [4]. Mutations in *MKKS* are found in 3–5% of families in which there are disease-causing mutations (Bardet–Biedl Syndrome or McKusick-Kaufman syndrome) [40,43]. Mutations in *BBS10* represent about 20% of all cases of BBS with certain ethnic variations: 43% in a Danish group and 8.3% in a Spanish cohort [44,45].

### 3.2. Mutational Heterogeneity

Mutational heterogeneity is important because testing strategies are consistent with mutation types. According to the Human Gene Mutation Database Professional (HGMD^®^) 2020.1 database (accessed in April 2021), 647 pathogenic variants are described. Figure 2 and Figure 3 show the implication of different types of genetic modifications in genes of BBSome, respectively the other genes implied in BBS. 

The gene of BBSome could present different types of mutations like: missense/nonsense, splicing substitutions, small deletions, gross deletions, small insertions/duplications, small indels, gross insertions/duplications, complex rearrangement. (Figure 2). For the other genes implied in BBS the most frequent type of mutations are: missense/nonsense, small deletions, and small insertions/duplications (Figure 3). [46,47,48]. Some BBS forms are produced by mutations with loss-of-function (LOF): nonsense, frameshift, copy number variants, and splicing variants [49].

BBSome is a multisubunit complex with eight proteins coding by the genes *BBS1*, *BBS2*, *BBS4*, *BBS5*, *BBS7*, *TTC8*, *BBS9* and *BBIP1* [13,35]. The most common mutations are in *BBS1* (responsible for 23% of BBS) and *BBS10* (identified in 20% of patients with BBS) [20]. Founder mutations have also been described: M390R in *BBS1* and C91fsX95 in *BBS10* [13]. *MKKS* is required for BBSome assembly, and *TTC8* is required for ciliary trafficking [50,51,52]. Mutations in the first 18 genes listed in the table occur in 70–80% of BBS affected families [53]. The *BBS1* and *BBS10* genes are the most common mutated genes in Europe and North America [54].

### 3.3. Modifiers

The phenotype of BBS can be changed by modifier genes. The PhenoModifier database contains 12 modifying variants described in 6 genes (*TMEM67*, *MKS1*, *MKKS*, *CCDC28B*-coiled-coil domain containing 28B, *C8orf37*, *BBS1*). Modifying genes influence expressivity or pleiotropy [55]. The pleiotropy is modified by some variants of *MKS1* gene: MKS1:c.1112_1114del (p.Phe371del), MKS1:c.1476T>G (p.Cys492Trp), MKS1:c.248A>G (p.Asp83Gly), MKS1:c.368G>A (p.Arg123Gln), MKS1:p.Ile450Thr in association with mutations in *BBS1* and *BBS10* genes [55,56]. All of these patients had seizures, non-specific sign for BBS or MKS (Meckel–Gruber Syndrome). Therefore, the interaction between the products of the two genes could explain this pleiotropic effect [56]. 

With an effect on expressivity we mention variants in *TMEM67*, *MKKS*, *CCDC28B*, *C8orf37*, *BBS1* genes [55,57,58,59]. For example, Bölükbaşı et al. found that most severe phenotype of BBS was allowed by following changes: C8orf37: c.533C>T (p.Ala178Val), CCDC28B: c.330C>T (p.Phe110Phe), MKKS: c.1015A>G (p.Ile339Val), and TMEM67: p.Asp799Asp [57]. Badano et al. showed that CCDC28B:c.430C/T variant (in heterozygote state) determines the introduction of a premature termination codon and the reduction of CCDC28B messenger RNA levels [58]. Badano et al. have identified another mutant alleles (BBS1: IVS115 + 2T→C) which is correlated with a more severe phenotype in a patient with homozygous mutation in *BBS2* gene [59].

### 3.4. Disease Protein–Non-Disease Protein Interconnectivity

There is functional interaction between complexes, which explains locus heterogeneity [8,60].

In an analysis conducted by Keith et al. in 2014, the authors showed the existence of multiple interconnectivity networks [61]. These relationships involve both the disease protein-disease protein connection, but also the disease protein–non disease protein. In the latter case, it appears that there are proteins which are related to more than one of the proteins encoded by the genes involved in BBS syndrome [61]. Examples of such proteins are: ALDOB (fructose-bisphosphate aldolase B) (in interaction with genes products *BBS1*, *BBS2*, *BBS4*, *BBS7*), HSCB (Iron-sulfur cluster co-chaperone protein HscB) (in interaction with genes products *BBS2*, *BBS4*, *BBS1*), ACY1 (Aminoacylase-1) (in interaction with genes products *BBS2*, *BBS4*, *BBS7*), EXOC7 (Exocyst complex component 7) (in interaction with genes products *BBS1*, *BBS7*, *BBS4*, *BBS2*), RAB3IP (Rab-3A-interacting protein) (in interaction with genes products *BBS1*, *BBS4*), FHOD1 (FH1/FH2 domain-containing protein 1) (in interaction with genes products *BBS1*, *BBS2*, *BBS4*, *BBS7*), PCM1 (Pericentriolar material 1 protein) in interaction with genes products *BBS1*, *BBS2*, *BBS4*, *BBIP1*), CCDC28B (Coiled-coil domain-containing protein 28B) (in interaction with genes products *MKKS*, *TTC8*, *BBS2*, *BBS4*, *BBS7*, *BBS1*, *BBS5*) [61].

Keith postulated that there is an interconnectivity between proteins in the case of locus heterogeneity. Thus, the BBS proteins in the vicinity of proteins involved in locus heterogeneity could be involved in the pathogenesis of similar diseases or may themselves be causes of disease [61].

The cellular signaling pathways involved in BBS are: hedgehog, Wnt (wingless-related integration site), GPCR (G protein-coupled receptors), mTOR (mammalian target of rapamycin) [5,6].

The primary cilium is involved in Hedgehog signaling pathway and can act as both positive and negative regulator of this pathway [5]. Defects in cilia/intraflagellar transport lead to loss of Hh phenotype function in the neural tube and gain of function in the limbs. This explains the common Hh phenotype (especially polydactyly) present in ciliopathies, including BBS [5,62].

Several BBS-associated proteins intervene in the Wnt signaling pathway and play a role in regulating Wnt signaling by degrading Wnt effectors [5,63]. Disheveled is an essential protein in basal body docking, ciliogenesis and planar cell polarity. Meckelin (TMEM67), TMEM216, MKS1, BBS10, and BBS12 also intervene in the same processes [5,51,64,65]. The lack of these proteins leads to planar cell polarity defects also found in BBS [5].

G-protein coupled receptors (GPCRs) are important for cilia structures and function. Neuronal cilia integrity is required for brain development and the adequate interaction in the adult brain [5]. Neuronal cilium formation begins prenatally with pro-cilium formation and continues in the first 8–12 weeks after birth. During postnatal development at the level of the primary neuronal membrane cilium GPCRs appear: somatostatin receptor 3 (SSTR3), melanin-concentrating hormone receptor 1 (MCHR1), serotonin receptor 6 (5HTR6), kisspeptin 1 receptor (KISS1R), dopamine receptors 1, 2, and 5 (D1, D2, and D5), neuropeptide Y receptors, NPY2R and NPY5R. The location of these receptors is different being consistent with the neuronal type. The length of the cilium is important for proper function. For example, the shortening of the cilia of hypothalamic neurons has been correlated with obesity in mice induced by high-fat diet, so that the dysfunction of neural cilia is correlated with childhood obesity in BBS [5,66,67]. GPCRs bind to b-arrestin2 and BBS proteins associated with Arl6 and thus trigger ciliary trafficking. When some of these receptors are missing or lacking motifs of recognition by BBSome and b-arrest, an accumulation of receptors and their ectocytosis occurs [68]. Rhodopsin and opsin are other examples of GPCRs in rods and cones. Their role is to absorb light and transmit the electrical signal to the brain. If the integrity of the cilia is not adequate, retinal degeneration will occur.

Mammalian target of rapamycin is also involved in BBS. CCDC28B interacts with the SN1 subunit of the mammalian target of rapamycin complex 2 (mTORC2). In this way it regulates the length of the cilia by affecting its assembly/stability [2].

### 3.5. Disease Protein-Disease Protein Interconnectivity

The eight subunits of BBSome are assembled in two parts: head and body. The head is formed by BBS2 which interacts with BBS7. The body is formed by the others proteins (BBS1, BBS4, BBS5, TTC8, BBS9, BBIP1). The BBIP1 is the central part of the BBSome core [68,69]. BBIP1 interacts with BBS4 and TTC8 and thus achieves proper assembly and structural stability of the BBSome complex [70]. BBS4 and TTC8 have TPR (tetratricopeptide repeats) subunits. They interact with the domain β propeller of BBS1 and BBS9, respectively [70]. These interactions are essential for a functional BBSome and explain why mutations at the BBS4-BBS1 and BBS8-BBS9 interface lead to disease in patients [70].

In a study that analyzed the effects of the proteome of the three modules (BBSome, chaperonin complex, transition zone) and three more genes that are not yet included in a module (*ARL6*, *TRIM32*, *WDPCP*) six epistatic interaction effects were identified: *BBS10-BBS5*; *BBS10-BBS12*; *BBS10-BBS1*; *BBS10-MKKS*; *BBS10-BBS4*; *BBS12-MKKS*. Moreover, 12 additive interaction effects were found between the main complexes (BBSome, chaperonin and transition zone) and 1 between proteins of transition zone and others gene *BBS1-BBS5*, *BBS1-BBS2*, *BBS1-NPHP1*, *BBS10-NPHP1*, *BBS2-NPHP1*, *BBS9-NPHP1*, *BBS7-NPHP1*, *BBS2-BBS4*, *BBS2-MKKS*, *BBS4-MKKS*, *BBS4-BBS9*, *BBS4-BBS7*. On the other hand, no interaction effect was demonstrated in the following genes (nor additive or epistatic effect): *TRIM32*, *WDPCP*, *CEP290*, *MKS1* [71].

## 4. Conclusions

Bardet–Biedl Syndrome is a rare ciliopathy characterized by a high degree of clinical heterogeneity. The mechanisms of heterogeneity are complex and explained by multiple loci, ubiquitous expression of genes, and multiple interaction of proteins coding by genes implicated in BBS. However, the deciphering of the entire mechanism implied in Bardet–Biedl Syndrome is a complex process that will require further research in the future.

## Figures and Tables

**Figure 1 genes-12-01353-f001:**
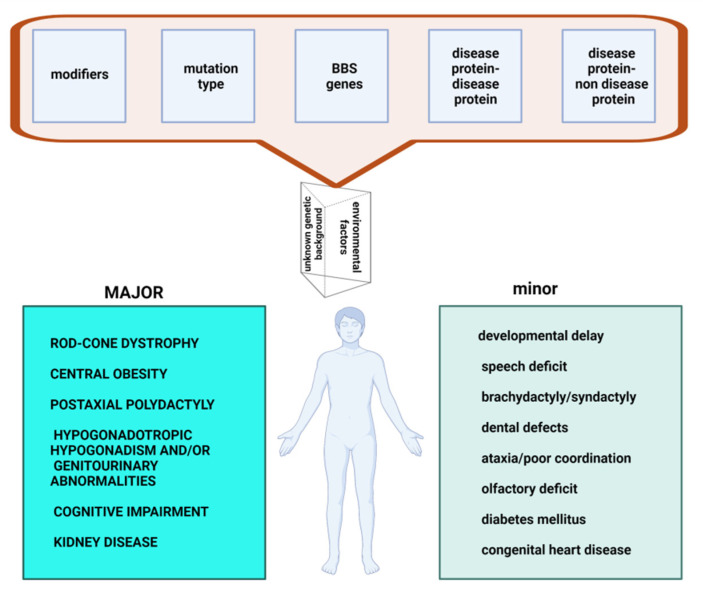
Heterogeneity in Bardet Biedl Syndrome-determinants and clinical effects. Created with BioRender.com (accessed on 15 August 2021).

**Figure 2 genes-12-01353-f002:**
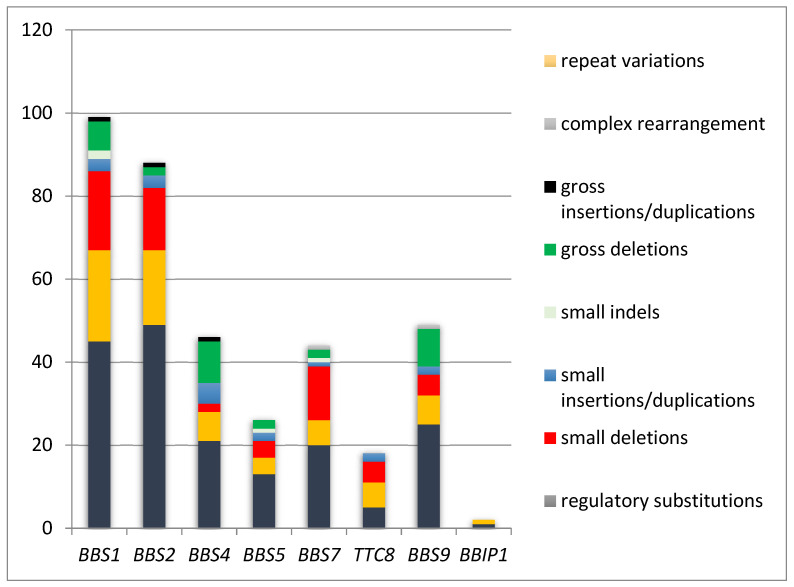
Pathogenic variants in BBSome [34]. *BBS1*: Bardet-Biedl syndrome 1; *BBS2*: Bardet-Biedl syndrome 2; *BBS4*: Bardet-Biedl syndrome 4; *BBS5*: Bardet-Biedl syndrome 5; *BBS7*: Bardet-Biedl syndrome 7; *TTC8*: tetratricopeptide repeat domain 8; *BBS9*: Bardet-Biedl syndrome 9; *BBIP1*: BBSome interacting protein 1.

**Figure 3 genes-12-01353-f003:**
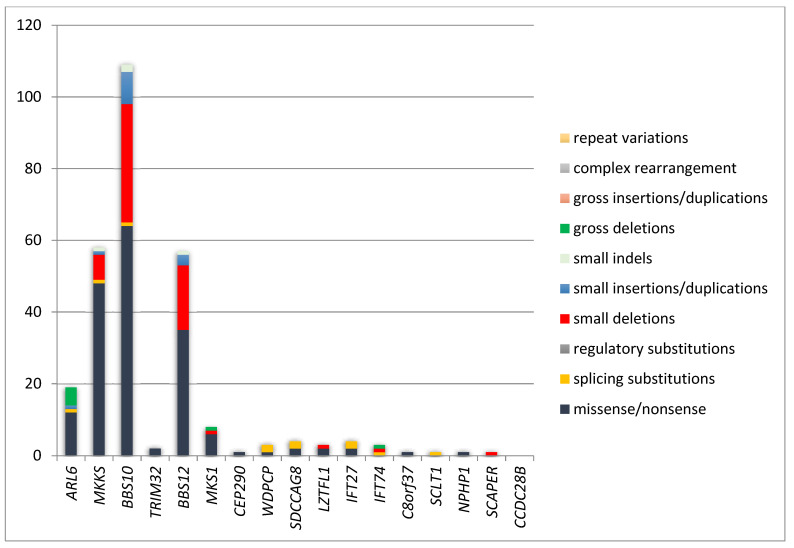
Pathogenic variants in other genes of BBS [34]. *ARL6*: ADP ribosylation factor like GTPase 6; *MKKS*: MKKS centrosomal shuttling protein; *BBS10*: Bardet-Biedl syndrome 10; *TRIM32*: tripartite motif containing 32; *BBS12*: Bardet-Biedl syndrome 12; *MKS1*: MKS transition zone complex subunit 1; *CEP290*: centrosomal protein 290; *WDPCP*: WD repeat containing planar cell polarity effector; *SDCCAG8*: SHH signaling and ciliogenesis regulator SDCCAG8; *LZTFL1*: leucine zipper transcription factor like 1; *IFT27*: intraflagellar transport 27; *IFT74*: intraflagellar transport 74; *C8orf37*: chromosome 8 open reading frame 37; *SCLT1*: sodium channel and clathrin linker 1; *NPHP1*: nephrocystin 1; *SCAPER:* S-phase cyclin A associated protein in the ER.

**Table 1 genes-12-01353-t001:** Genes in Bardet–Biedl Syndrome: characteristics, tissue, and single cell type specificity [34,35,36,37].

No	Gene Symbol	Gene Name	Gene Groups	Chromosome	Protein	Localization	Tissue Specificity
1	*BBS1*	Bardet–Biedl syndrome 1	BBSome	11q13.2	Bardet–Biedl syndrome 1 protein	Basal body, cilium	Low tissue specificity
2	*BBS2*	Bardet–Biedl syndrome 2	BBSome	16q13	Bardet–Biedl syndrome 2 protein	Basal body, cilium	Low tissue specificity
3	*ARL6*	ADP ribosylation factor like GTPase 6	ARF GTPase family	3q11.2	ADP-ribosylation factor-like protein 6	Basal body, cilium, cytosol, transition zone	Low tissue specificity
4	*BBS4*	Bardet–Biedl syndrome 4	Tetratricopeptide repeat domain containing BBSome	15q24.1	Bardet–Biedl syndrome 4 protein	Basal body, cilium	Low tissue specificity
5	*BBS5*	Bardet–Biedl syndrome 5	BBSome	2q31.1	Bardet–Biedl syndrome 5 protein	Basal body	Low tissue specificity
6	*MKKS*	MKKS centrosomal shuttling protein	Chaperonins	20p12.2	McKusick–Kaufman/Bardet–Biedl syndromes putative chaperonin	Basal body, cytosol	Low tissue specificity
7	*BBS7*	Bardet–Biedl syndrome 7	BBSome	4q27	Bardet–Biedl syndrome 7 protein	Basal body, cilium	Low tissue specificity
8	*TTC8*	Tetratricopeptide repeat domain 8	Tetratricopeptide repeat domain containing BBSome	14q31.3	Tetratricopeptide repeat protein 8	Basal body, cilium, IFT	Low tissue specificity
9	*BBS9*	Bardet–Biedl syndrome 9	BBSome	7p14.3	Protein PTHB1	Cilium	Low tissue specificity
10	*BBS10*	Bardet–Biedl syndrome 10	Chaperonins	12q21.2	Bardet–Biedl syndrome 10 protein	Basal body	Low tissue specificity
11	*TRIM32*	Tripartite motif containing 32	Tripartite motif containing Ring finger proteins	9q33.1	E3 ubiquitin-protein ligase TRIM32	Intermediate filaments	Low tissue specificity
12	*BBS12*	Bardet–Biedl syndrome 12	Chaperonins	4q27	Bardet–Biedl syndrome 12 protein	Basal body	Low tissue specificity
13	*MKS1*	MKS transition zone complex subunit 1	B9 domain containing MKS complex	17q22	Meckel syndrome type 1 protein	Basal body	Low tissue specificity
14	*CEP290*	centrosomal protein 290	MKS complex	12q21.32	Centrosomal protein of 290 kDa (Cep290)	Basal body, centrosome	Low tissue specificity
15	*WDPCP*	WD repeat containing planar cell polarity effector	Ciliogenesis and planar polarity effector complex subunits	2p15	WD repeat-containing and planar cell polarity effector protein fritz homolog (hFRTZ)	Cytosol, plasma membrane, axoneme	Low tissue specificity
16	*SDCCAG8*	SHH signaling and ciliogenesis regulator SDCCAG8	MicroRNA protein coding host genes	1q43-q44	Serologically defined colon cancer antigen 8	Basal body, centriole, transition zone	Low tissue specificity
17	*LZTFL1*	leucine zipper transcription factor like 1		3p21.31	Leucine zipper transcription factor-like protein 1	Basal body, cilium	Tissue enhanced (lymphoid tissue)
18	*BBIP1*	BBSome interacting protein 1	BBSome	10q25.2	BBSome-interacting protein 1	Cytoplasm, cytosol	Tissue enhanced (testis)
19	*IFT27*	intraflagellar transport 27	IFT-B1 complex RAB, member RAS oncogene GTPases	22q12.3	Intraflagellar transport protein 27 homolog	Basal body, cilium, IFT	Low tissue specificity
20	*IFT74*	intraflagellar transport 74	IFT-B1 complex	9p21.2	Intraflagellar transport protein 74 homolog	Basal body, cilium, IFT	Low tissue specificity
21	*C8orf37*	chromosome 8 open reading frame 37		8q22.1	Protein C8orf37	Basal body, ciliary root	Low tissue specificity
22	*SCLT1*	sodium channel and clathrin linker 1		4q28.2	Sodium channel and clathrin linker 1	Centriole	Low tissue specificity
23	*NPHP1*	Nephrocystin 1	NPHP complex	2q13	Nephrocystin-1	Transition zone	Tissue enhanced (skeletal muscle)
24	*SCAPER*	S-phase cyclin A associated protein in the ER	Zinc fingers C2H2-type	15q24.3	S phase cyclin A-associated protein in the endoplasmic reticulum (S phase cyclin A-associated protein in the ER)	Endoplasmic reticulum	Low tissue specificity

## Data Availability

Some data were obtained from Human Gene Mutation Database (HGMD^®^ Professional 2021.1) available online: http://www.hgmd.cf.ac.uk/ac/all.php, accessed on 1 April 2021.

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
