# Peer review of "Bardet–Biedl Syndrome—Multiple Kaleidoscope Images: Insight into Mechanisms of Genotype–Phenotype Correlations"

_genes, 2021, doi:10.3390/genes12091353_

Round 1

Reviewer 1 Report

An up-to-date review of the genetics of Bardet-Biedl syndrome would be useful to the field.  However, in its current form this review falls short in several ways.

First, the descriptions tend to be superficial without sufficient depth of analysis, especially in the modifiers and connectivity paragraphs (sections 3.3-3.5).

Second, the descriptions tend to repeat observations from previous reviews without providing new insight.  This issue was obvious in the chaperonins section on p.8.

Third, the tables and figures could be presented more clearly.  For example, in Table 1 the cell specificity column on the right has too much vertical text, making it difficult to read.  In Figures 2 and 3, the colors are too similar to clearly distinguish between the different types of variants.

Fourth, the English writing contains a number of poorly worded sentences and grammatical errors.

Author Response

We thank the reviewer for giving us the opportunity to improve the quality of the manuscript.

“An up-to-date review of the genetics of Bardet-Biedl syndrome would be useful to the field.  However, in its current form this review falls short in several ways.

First, the descriptions tend to be superficial without sufficient depth of analysis, especially in the modifiers and connectivity paragraphs (sections 3.3-3.5).”

Answer: We rewrote and added information in the modifiers and connectivity paragraphs.

“Second, the descriptions tend to repeat observations from previous reviews without providing new insight.  This issue was obvious in the chaperonins section on p.8.”

Answer: We added and completed the section 3. Determinants of clinical effects.

“Third, the tables and figures could be presented more clearly.  For example, in Table 1 the cell specificity column on the right has too much vertical text, making it difficult to read.  In Figures 2 and 3, the colors are too similar to clearly distinguish between the different types of variants.”

Answer: We changed the colours of the figures 2 and 3. We deleted the last column in table 1, but the information is in the text.

“Fourth, the English writing contains a number of poorly worded sentences and grammatical errors.”

Answer: We revised the manuscript and did language revisions.

Reviewer 2 Report

Thank you for the opportunity to review the manuscript “Bardet-Biedl Syndrome – multiple kaleidoscope images: insight into mechanisms of genotype-phenotype correlations” by Laura Florea et al. Bardet-Biedl Syndrome (BBS) is a rare, genetically and clinically heterogeneous disorder. Therefore, I find the review interesting and valuable for both geneticists and clinicians. 

However, there are some points that need to be improved.

Major comments:

  1. In my opinion there should be a paragraph summarizing the diagnostic process (clinical, genetical). There is lack of describing the genetic methods that can be used in the diagnostic process. It seems to be important for clinicians taking care of the patients with BBS.
  2. Fig 1 is unclear. There is lack of describing which features are major and which minor? What are environmental factors?
  3. Lines 63-68- “primary” and “secondary”- shouldn’t it be rather major and minor?
  4. Lines 75-76, 97-99- the meaning of the text is unclear, please modify.
  5. Line 97- please explain what do you mean by “complete form of hypogonadism”?
  6. There is lack of important references: lines 99, 112-118, 128-131, 136-139.
  7. Line 127- please explain what are “unspecified heart anomalies”.
  8. Line 135- “subclinical hypothyroidism”- please explain in more detail.
  9. Line 188 and following- genes’ names should be in italics
  10. English language needs extensive editing

Minor comments:

  1. Lines 27-28, 35, 89, 181- lack of explaining the abbreviations.
  2. Line 42- “motor”- shouldn’t it be motile?
  3. Line 144- Bardet-Biedl syndrome- please use BBS.
  4. Fig 2 and Fig 3- lack of explaining the abbreviations.
  5. Lines 241-242- need editing.

Author Response

We thank the reviewer for giving us the opportunity to improve the quality of the manuscript.  

Major comments:

  1. “In my opinion there should be a paragraph summarizing the diagnostic process (clinical, genetical). There is lack of describing the genetic methods that can be used in the diagnostic process. It seems to be important for clinicians taking care of the patients with BBS.”

Answer: We introduced a paragraph with the diagnostic process.

  1. “Fig 1 is unclear. There is lack of describing which features are major and which minor? What are environmental factors?”

Answer: We explained in the figure which features are majore and which minor.

  1. “Lines 63-68- “primary” and “secondary”- shouldn’t it be rather major and minor?”

Answer: We changed “primary” and “secondary” with “major” and “minor”.

  1. “Lines 75-76, 97-99- the meaning of the text is unclear, please modify.”

Answer: We changed.

  1. “Line 97- please explain what do you mean by “complete form of hypogonadism”?”

Answer: We replaced “complete form of hypogonadism” with “hypogonadotropic hypogonadism”.

  1. “There is lack of important references: lines 99, 112-118, 128-131, 136-139.”

Answer: We completed the references.

  1. “Line 127- please explain what are “unspecified heart anomalies”.

Answer: We changed the phrase. It’s about the frequency of cardiovascular anomalies, not a specific anomaly.

  1. “Line 135- “subclinical hypothyroidism”- please explain in more detail.”

Answer: We explained “Subclinical hypothyroidism is a mild form of hypothyroidism characterized  by peripheral thyroid hormone levels within normal reference laboratory range but serum thyroid-stimulating hormone (TSH) levels are mildly elevated”

  1. “Line 188 and following- genes’ names should be in italics”

Answer: We wrote in italis.

  1. “English language needs extensive editing”

Answer: We revised the manuscript and did language revisions.

Minor comments:

  1. “Lines 27-28, 35, 89, 181- lack of explaining the abbreviations”.

Answer: We explained the abbreviations.

  1. “Line 42- “motor”- shouldn’t it be motile?”

Answer: We replaced “motor” with “motile”

  1. “Line 144- Bardet-Biedl syndrome- please use BBS.”

Answer: We used BBS.

  1. “Fig 2 and Fig 3- lack of explaining the abbreviations.”

Answer: We explained the abbreviations.

  1. “Lines 241-242- need editing.”

Answer: We made the editing.

We thank the reviewer for giving us the opportunity to improve the quality of the manuscript.  

Major comments:

  1. “In my opinion there should be a paragraph summarizing the diagnostic process (clinical, genetical). There is lack of describing the genetic methods that can be used in the diagnostic process. It seems to be important for clinicians taking care of the patients with BBS.”

Answer: We introduced a paragraph with the diagnostic process.

  1. “Fig 1 is unclear. There is lack of describing which features are major and which minor? What are environmental factors?”

Answer: We explained in the figure which features are majore and which minor.

  1. “Lines 63-68- “primary” and “secondary”- shouldn’t it be rather major and minor?”

Answer: We changed “primary” and “secondary” with “major” and “minor”.

  1. “Lines 75-76, 97-99- the meaning of the text is unclear, please modify.”

Answer: We changed.

  1. “Line 97- please explain what do you mean by “complete form of hypogonadism”?”

Answer: We replaced “complete form of hypogonadism” with “hypogonadotropic hypogonadism”.

  1. “There is lack of important references: lines 99, 112-118, 128-131, 136-139.”

Answer: We completed the references.

  1. “Line 127- please explain what are “unspecified heart anomalies”.

Answer: We changed the phrase. It’s about the frequency of cardiovascular anomalies, not a specific anomaly.

  1. “Line 135- “subclinical hypothyroidism”- please explain in more detail.”

Answer: We explained “Subclinical hypothyroidism is a mild form of hypothyroidism characterized  by peripheral thyroid hormone levels within normal reference laboratory range but serum thyroid-stimulating hormone (TSH) levels are mildly elevated”

  1. “Line 188 and following- genes’ names should be in italics”

Answer: We wrote in italis.

  1. “English language needs extensive editing”

Answer: We revised the manuscript and did language revisions.

Minor comments:

  1. “Lines 27-28, 35, 89, 181- lack of explaining the abbreviations”.

Answer: We explained the abbreviations.

  1. “Line 42- “motor”- shouldn’t it be motile?”

Answer: We replaced “motor” with “motile”

  1. “Line 144- Bardet-Biedl syndrome- please use BBS.”

Answer: We used BBS.

  1. “Fig 2 and Fig 3- lack of explaining the abbreviations.”

Answer: We explained the abbreviations.

  1. “Lines 241-242- need editing.”

Answer: We made the editing.

Round 2

Reviewer 2 Report

Please see the attached review.

Author Response

Thank you very much for this comment on the paper.

Yes, the autoimmune thyroid disorder in BBS is more prevalent that in general population. We added a phrase with this information

“Hashimoto's thyroiditis is the most common form of subclinical hypothyroidism and is more prevalent in BBS patients compared to the general population.”